# TAFRO Syndrome on ^18^F-FDG-PET/CT: An Appealing Diagnostic Tool

**DOI:** 10.3390/diagnostics14101025

**Published:** 2024-05-16

**Authors:** Ayoub Jaafari, Nadim Taheri, Sohaïb Mansour, Saïf-Eddine El Bouhali, Rachid Attou

**Affiliations:** 1Department of Nuclear Medicine, Brussels University Hospital (H.U.B), 1070 Brussel, Belgium; 2Department of Rheumatology, C.H.U Brugmann, 1020 Brussel, Belgium; nadim.taheri@gmail.com; 3Department of Internal Medicine, C.H.U Brugmann, 1020 Brussel, Belgium; sohaib.mansour@chu-brugmann.be; 4Department of Intensive Care Unit, C.H.U Brugmann, 1020 Brussel, Belgium; saif.elbouhali@gmail.com (S.-E.E.B.); rachid.attou@chu-brugmann.be (R.A.)

**Keywords:** TAFRO syndrome, multicentric Castleman disease, iMCD, 18F-FDG-PET/CT

## Abstract

TAFRO syndrome (TS) is a recently recognized and heterogenous systemic disease characterized by a confluence of symptoms: thrombocytopenia (T), anasarca (A), fever (F), reticulin myelofibrosis (R), and organomegaly (O). First described in Japan in 2010, the pathogenesis remains unclear and includes various clinical conditions such as malignancies, rheumatologic disorders, infections, and “Polyneuropathy, Organomegaly, Endocrinopathy, Monoclonal plasma cell disorder, and Skin changes” (POEMS) syndrome. Due to its heterogeneous presentation and potential life-threatening delays in diagnosis, accurate diagnosis is crucial. According to the literature, no specific imaging modality has been recommended for the work-up of patients with suspected TS. Here, we report a case of TS and its management using 18F-FDG-PET/CT imaging as an attractive complementary diagnostic tool.

**Figure 1 diagnostics-14-01025-f001:**
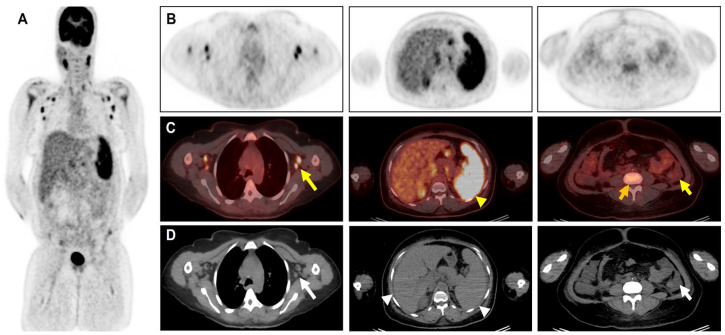
^18^F-FDG-PET/CT of the patient: (**A**) Maximum Intensity Projection (MIP) image of the chest, and upper and lower abdomen, respectively (left to the right), (**B**) PET images, (**C**) fused PET/CT images, (**D**) CT images. A 44-year-old Mauritanian woman without significative pre-existing medical and recent traveling history presented with tiredness and febrile non-bloody diarrhea for three weeks. No other complaints. Physical examination revealed left cervical adenopathy, anasarca, and mild splenomegaly. Laboratory tests showed non-regenerative anemia (5.5 g/L), hypoplakettosis (23,000/µL), elevated serum concentration of C-reactive protein (121 mg/dL), direct positive Coombs test without acute hemolysis, and no hypergammaglobulinemia. Extensive work-up for infections (viral, bacterial, and fungal) and autoimmune diseases were also negative. A thoracoabdominal scanner was realized and showed slightly enlarged cervicothoracic adenopathy (large white arrow), hepatosplenomegaly (head white arrow), and ascites (small white arrow). The bone marrow was initially explored via a sternal marrow puncture showing a dry tap, with inconclusive results (Figure 2A). An ensuing bone marrow biopsy was performed, and exposed reticulin myelofibrosis, and numerous megakaryocytes bordering dilated sinusoids (Figure 2B). Metabolic imaging with ^18^F-FDG-PET/CT was achieved to assess underlying causes and identify the most active lymph node for biopsy. MIP (**A**) and fused images show increased and diffuse FDG uptake in the cervicothoracic lymph nodes (large yellow arrow), spleen (yellow head arrow) ascites (small yellow arrow), and vertebral medulla (orange arrow). Lymph node biopsy revealed rare follicles, atrophy of germinal centers, proliferation, hyperplasia of blood vessels, and plasma cells suggestive of Castleman’s disease (Figure 2C,D). Tests for herpes human virus-6 (HHV-6), HHV-8, and human immunodeficiency virus (HIV) were negative. These results, alongside the biological, CT, and ^18^F-FDG PET/CT findings, suggest a diagnosis of TAFRO syndrome (TS). She was successfully treated with corticosteroid and immunotherapy (interleukin-6 inhibitors) with a clinical and biological improvement. First described by Takai et al. in 2010, TAFRO syndrome (TS) is a rare inflammatory lymphoproliferative disease characterized by a highly heterogeneous clinical presentation involving thrombocytopenia (T), anasarca (A), fever (F), reticulin fibrosis (R), and organomegaly (O), and it is potentially life-threatening [1]. While the majority of reported TS cases have originated in Japan, it has increasingly been recognized in non-Asian patients worldwide [2]. TS includes various clinical conditions such as malignancies, rheumatologic disorders, infections, and “Polyneuropathy, Organomegaly, Endocrinopathy, Monoclonal plasma cell disorder, and Skin changes” (POEMS) syndrome [3]. To date, the most influential diagnostic criteria for TS were proposed by Iwaki et al. [4] and Masaki et al. [5] in 2016, and Masaki et al. [6] updated the diagnostic criteria in 2019. Patients with TS commonly present with more-severe clinical symptomatology and worse outcomes than those with iMCD due to a greater cytokine storm fostering a pro-inflammatory (potentially life-threatening) state, explaining the clinical symptomatology and laboratory abnormalities, such as anemia, thrombocytopenia, elevated C-reactive protein (CRP), and the absence of hyper-gammaglobulinemia, as found in our patient, which is a distinct feature of TS [7]. According to existing literature, no specific imaging modality has been universally recommended for the evaluation of suspected iMCD or TS cases. Typically, computerized tomography (CT) scanning reveals nonspecific multiple lymphadenopathies that may indicate other differential diagnoses such as lymphoproliferative disorders or granulomatosis [8]. However, this is not always the situation. Certain cases of TS have been described in the literature where adenopathy was not necessarily found [9]. Moreover, early diagnosis of TS on imaging may sometimes point to the aggressiveness of the disease and be an unfavorable prognostic indicator. In addition to pleural and abdominal effusions and lymphadenopathy, some authors describe the presence of other abnormalities such as adrenal abnormalities (hemorrhage, ischemia, adenoma) in the early stages of TS, which were not found in our patient [10]. In metabolic imaging, management with 2-[18F]-fluoro-2-deoxy-D-glucose (18F-FDG) positron emission tomography combined with computed tomography (PET/CT) is mainly used to identify high-uptake lesions. This hybrid imaging method provides information on metabolic characteristics and anatomical location. Consequently, determining the highest metabolic activity in mild or enlarged lymph nodes potentially facilitates the identification of suitable biopsy targets, leading to earlier diagnosis. Moreover, certain disease features like myelofibrosis are challenging to detect on CT imaging alone, but 18F-FDG-PET/CT can demonstrate diffuse uptake in the marrow (orange arrow), capturing key findings of TS [11,12]. Additionally, uptake in hepatosplenomegaly provides supplementary information by reflecting sites of secondary hematopoiesis, aiding in diagnosis [11]. Identifying the cause of ascites poses a challenge due to its association with various pathologies, ranging from non-malignant conditions like chronic liver disease and heart failure to malignant neoplasms [13]. In our case, observing hypermetabolism in ascites is particularly intriguing. While ascites are generally present in all TS cases, different studies have described varying sensitivity and specificity values of 18F-FDG-PET/CT for its characterization [13,14]. Furthermore, combining metabolic imaging with tumor biomarkers such as interleukin-6 (IL-6) and vascular endothelial growth factor (VEGF) levels in effusions and serum can provide valuable insights for investigating TS [13,14]. To date, only a few TS cases have been reported, and the utilization of 18F-FDG-PET/CT as a complementary diagnostic tool remains limited [15,16,17]. Given the heterogeneous presentation of the disease and potential life-threatening delays in diagnosis, accurate diagnosis is crucial to initiate appropriate treatment. In conclusion, TAFRO syndrome is a heterogeneous disease that involves a pattern of symptoms and may be potentially life threatening if not diagnosed early. Futhermore, 18F-FDG-PET/CT holds promise as a complementary tool for providing characteristic findings of TS. We suggest that the diagnosis should be considered when a patient presents with a combination of clinical (fever), biological (anemia, thrombocytopenia, absence of polyclonal hypergammaglobulinemia), and imaging findings such as the presence of effusion (pleural, associated or not with FDG uptake; diffuse FDG uptake in the bone marrow; hepatosplenomegaly with high splenic FDG uptake; and hypermetabolic lymph nodes).

**Figure 2 diagnostics-14-01025-f002:**
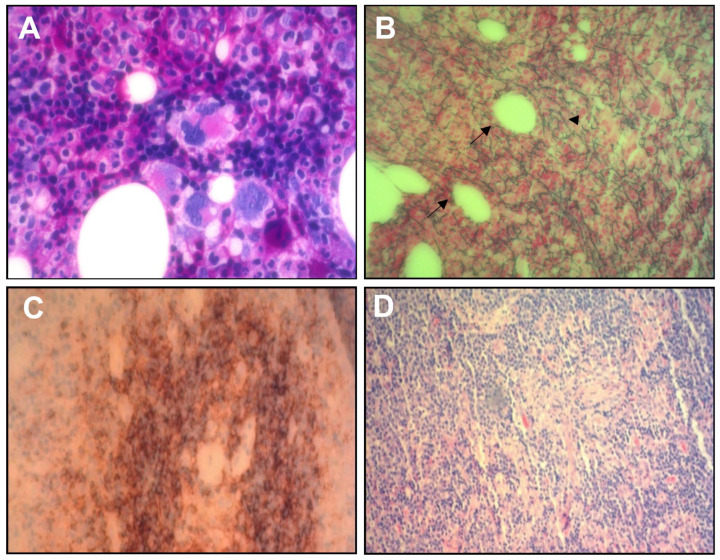
(**A**) 100× hematoxylin and eosin stain of the marrow puncture showing a “dry tap” characterized by slightly hypercellular marrow with increased megakaryocytes. (**B**) Reticulin stain of bone marrow biopsy showed reticulin myelofibrosis (head arrow, green fibers), and numerous megakaryocytes bordering dilated sinusoids (arrow). Lymph node biopsy showed rare follicles, atrophy of germinal centers (**C**), proliferation, and hyperplasia of blood vessels (**D**).

## Data Availability

The data used and analyzed in this study are available from the corresponding author on reasonable request.

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
