# Peer review of "TAFRO Syndrome on 18F-FDG-PET/CT: An Appealing Diagnostic Tool"

_diagnostics, 2024, doi:10.3390/diagnostics14101025_

Round 1
Reviewer 1 Report
Comments and Suggestions for Authors
This report is interesting, however author must alter some description about TAFRO syndrome.
1, Author described that TAFRO recognized as a subtype of MCD in abstract
and introduction, but this description is wrong.
Because some TAFRO syndrome case do not demonstrate lymphadenopathy, thus iMCD and TAFRO syndrome may overlap but TAFRO do not included into iMCD. iMCD-TAFRO may be a part of iMCD.
Author Response
- Yes I'm agree with your text event the exact pathogenesis remains unclear it is also referred to as a severe variant of idiopathic multisystemic Castleman's disease (iMCD) in international treatment guidelines. Additionally, TS includes various clinical conditions such as malignancies, rheumatologic disorders, infections, and (POEMS). To date, the most influential diagnostic criteria for TAFRO syndrome were proposed by Iwaki et al. and Masaki et al. respectively in 2016, and Masaki et alupdated the diagnostic criteria in 2019 according to its clinical, biological, and histopathological features.
- We have made the appropriate changes by explaining the concept of TAFRO much more clearly.
Reviewer 2 Report
Comments and Suggestions for Authors
This is an interesting case presented with 18F-FDG PET/CT findings. However, the discussion should be further improved.
1. In this case, pleural fluid and ascites were low for TAFRO syndrome. This may be due to the fact that the imaging was performed early in the course of the disease (PMID: 32440781).
2. It has also been noted that adrenal abnormalities may be a characteristic finding in TAFRO syndrome, if present, that is not present in non-TAFRO-iMCD (https://doi.org/10.3390/biomedicines12040837). Were there any abnormal findings in the adrenal glands, including FDG uptake?
Author Response
1. In our case, the patient was at a very early stage of her disease and therefore the pleural and abdominal effusion did not appear to be very significant. Fortunately, she was diagnosed very early, but this is probably due to the early onset of TAFRO symptoms.
2. Standard conventional imaging showed no adrenal infiltration or other associated abnormalities (haemorrhage, ischaemia, whatever). On 18FDG-PET/CT, there was no adrenal abnormality in our patient. In addition, no adrenal abnormality could be identified in the clinico-biological and radiological evolution. (rapid management of the disease vs TAFRO syndrome without adrenal abnormality).
We already had enough clinical, biological and imaging diagnostic elements/criteria (organomegaly, pleural and abdominal effusion, adenomegaly/pathology) as elements of poor prognosis at an early stage.